# Chronic pain precedes disrupted eating behavior in low-back pain patients

**Yezhe Lin[1], Ivan De Araujo[2], Gelsina Stanley[3,4], Dana Small[3,4], Paul Geha[1,3,4]***

**1** Department of Psychiatry, School of Medicine and Dentistry, University of Rochester, Rochester, New York, United States of America, **2** Neuroscience Department, Icahn School of Medicine at Mount Sinai, New York, New York, United States of America, **3** The John B. Pierce Laboratory, New Haven, Connecticut, United States of America, **4** Department of Psychiatry, Yale School of Medicine, Yale University, New Haven, Connecticut, United States of America

* paul_geha@urmc.rochester.edu

**Data Availability Statement:** All relevant data are within the manuscript and its Supporting Information files.

**Funding:** National Institute on Drug Abuse 5K08DA037525 Dr. Paul Geha Department of

## Abstract

Chronic pain is associated with anhedonia and decreased motivation. These behavioral alterations have been linked to alterations in the limbic brain and could explain the increased risk for obesity in pain patients. The mechanism of these behavioral changes and how they set in in relation to the development of chronic pain remain however poorly understood. Here we asked how eating behavior was affected in low-back pain patients before and after they transitioned to chronic pain, compared to patients whose pain subsided. Additionally, we assessed how the hedonic perception of fat-rich food, which is altered in chronic pain patients, related to the properties of the nucleus accumbens in this patients' population. We hypothesized that the accumbens would be directly implicated in the hedonic processing of fat-rich food in pain patients because of its well-established role in hedonic feeding and fat ingestion, and its emerging role in chronic pain. Accordingly, we used behavioral assays and structural brain imaging to test sub-acute back pain patients (SBP) and healthy control subjects at baseline and at approximately one-year follow-up. We also studied a sample of chronic low-back pain patients (CLBP) at one time point only. We found that SBP patients who recovered at follow-up (SBPr) and CLBP patients showed disrupted eating behaviors. In contrast, SBP patients who persisted in having pain at follow-up (SBPp) showed intact eating behavior. From a neurological standpoint, only SBPp and CLBP patients showed a strong and direct relationship between hedonic perception of fat-rich food and nucleus accumbens volume. This suggests that accumbens alterations observed in SBPp patients in previous works might protect them from hedonic eating disruptions during the early course of the illness. We conclude that disrupted eating behavior specifically sets in after pain chronification and is accompanied by structural changes in the nucleus accumbens.

## Introduction

How chronic pain affects eating behavior and body weight is still poorly understood. The premise that appetitive drives such as eating and chronic pain are mechanistically linked is

Psychiatry at the Yale School of Medicine Dr. Paul Geha School fo Medicine and Dentistry, University of Rochester Dr. Paul Geha The Del Monte Neuroscience Institute Dr. Paul Geha The Funders had no role in study design, data collection and analysis, decision to publish or preparation of the manuscript.

**Competing interests:** The authors have declared that no competing interests exist.

grounded in the fact that both pain and eating interact with motivational states and affect decision making [1–5]. This hypothetical mechanistic link is of high clinical significance since chronic pain and obesity are often comorbid [6,7]. In fact, chronic pain is more prevalent in obese individuals and obesity is highly prevalent in different types of chronic pain [8–10].

According to the "fear-avoidance model," pain patients may become obese due to lack of sufficient physical activity. Specifically, chronic pain leads to aversion to movement in pain patients as a way to avoid exacerbating their pain, and later to "deconditioning", and gradually patients become intolerant of physical activity [11]; However, this hypothesis has been met with underwhelming evidence, as previous studies have failed to detect significant changes in physical activity levels in pain patients (as measured by questionnaires or accelerometers) [12–17].

An alternative explanation is that chronic pain and obesity interact at the level of the central nervous system. The brain plays a key role in the regulation of energy intake and expenditure [18], with recent findings emphasizing the role of reward limbic circuits in the current obesity epidemic [19,20]. According to this explanation, interindividual differences in brain response to food cues [3,19] interact with a modern diet environment rich in fats and carbohydrates [21], resulting in excessive caloric intake in segments of the population [22]. On the other hand, recent pre-clinical and neuroimaging studies have demonstrated that chronic pain is associated with neuroadaptations in the brain limbic system [23], and that the risk of transition to chronic pain can be predicted from the structural and functional properties of the limbic system [24–26]. Consistently, both chronic pain and obesity are characterized by anhedonia [27–30], hypo-dopaminergic state in the mesocorticolimbic system [31–36], and altered opioid transmission within the limbic system [37,38].

Building on this overlap, we have previously shown [17] that chronic low back pain (CLBP) patients exhibit decreased liking of high-fat puddings but not of high sucrose drinks, suggesting an overlap between circuits underlying CLBP and fat valuation. More importantly, we showed that liking predicted caloric intake from a high-fat pudding offered *ad-libitum* in healthy controls but not in CLBP patients, indicating the presence of disrupted satiety signals in the latter group [17]. Here we asked how eating behavior was affected in low-back pain patients before and after they recovered or transitioned to chronic pain. Therefore, we studied the eating behavior of sub-acute low-back pain patients and healthy controls at baseline and at long-term follow-up after almost one year. Given the role of the nucleus accumbens (NAc) in the hedonic perception of highly palatable food and subjective liking ratings [39–41], we also tested whether NAc volume predicted hedonic ratings and/or caloric intake in our participants. This analysis tested our overall hypothesis of the direct relationship between limbic brain changes in low-back pain and disrupted eating behavior.

## Methods

### Subjects

All subjects gave written informed consent to participate in the study, which was approved by the Yale University Institutional Review Board. We received > 500 responses to our ads from patients complaining of back pain. 68 chronic low-back pain (CLBP) patients completed the full phone screening, 43 (17 males) of whom participated in the study; 97 SBP subjects completed the full phone screening, 51 (31 males) of whom participated in the study; 79 healthy controls were screened, 36 of whom participated in our study. The study consisted of two time points separated by approximately one year (baseline, 1-year follow-up). At each time point, participants completed 2 sessions (**Fig 1**). 36 healthy controls (HC) (20 males), 51 SBP subjects, and 43 chronic low back pain (CLBP) patients completed the baseline visits and finished

**Fig 1. Schematic diagram showing the experiments performed at both time-points and during each session.** Note that CLBP patients were tested once only, while SBP patients and healthy subjects were tested at two time points separated by approximately one year (i.e., baseline and follow-up).

session 1 experiments. Of the SBP group, 20 patients were confirmed at follow-up as recovered subacute back pain patients (SBPr) and 16 as persistent subacute back pain patients (SBPp). Patients whose pain dropped by more than 30% [42] at follow-up were considered recovered, otherwise persistent. 41 SBP, 33 CLBP and 30 healthy subjects finished experiments of session 2 at baseline, 20 of whom were SBPr and 16 were SBPp. At follow-up roughly 1 year from baseline, the same experiments were repeated. 16 HC and 29 SBP (16 SBPr and 13 SBPp) subjects completed session 1. 15 SBPr, 12 SBPp and 15 HC subjects completed session 2. Subjects were recruited through flyers in the New Haven area and advertisements on the internet. Subjects were briefly screened at first to check (1) the location of back-pain, (2) if they were otherwise healthy, (3) if they were non-smokers, and (4) their pain duration (between 6 and 12 weeks for SBP and more than one year for CLBP). If they passed this initial brief screen, a more detailed screen was conducted where we assessed demographics, location, possible cause, duration, radiation of the pain, analgesic medication use, medical work-up of the back-pain, substance misuse, recent or past history of opioid medication use, complete medical and psychiatric history, recent or past fluctuations in body weight, history of olfactory or taste impairments, or nasal sinuses surgery. To be included in the study, SBP subjects needed to meet the criteria of having a new-onset 6 to12 weeks low-back pain with no back pain in the year prior to the ongoing episode; CLBP patients had to have a pain duration of at least 1 year. Both SBP and CLBP participants had to (1) fulfill the International Association for the Study of Pain criteria for back pain [43], (2) not be currently, or during the month prior to the study, on any opioid analgesics. We chose to include CLBP patients with at least 1 year of back pain to ensure that they were in the time window when brain reorganization had set in [26,44]. Patients were included if their back pain was below the T-12 thoracic vertebra with or without radiculopathy and was present on more days than not. SBP and CLBP diagnoses were confirmed based on history collected by an experienced clinician (P.G.). Healthy control subjects were screened likewise with the additional exclusion criteria of any current back pain or any history of any pain of more than 6 weeks in duration as an exclusion criterion. Participants had no history of psychiatric disorders, chronic medical conditions (e.g., diabetes, coronary artery disease), loss of consciousness, chemosensory impairment, or food allergies.

## Stimuli

A set of four pudding samples were prepared with 0%, 3.1%, 6.9%, and 15.6% fat content weight by weight (w/w) [45]. The samples were prepared by mixing instant pudding in heavy cream or milk (Guida's Dairy) with varying fat content. The sugar content was held constant between the four stimuli at 4.6% (w/w). To maximize liking (i.e., hedonic) ratings, subjects were asked to pick a preferred flavor from a choice of vanilla and chocolate during the pre-study screening interview. A set of four orange or strawberry-flavored Jell-Os with 0, 0.1, 0.56, and 1 Molar (M) sucrose concentration respectively were prepared and represented the sugar stimuli.

## Procedures

Fig 1 is a schematic diagram describing the procedures done and the groups undergoing these procedures at baseline and at long-term follow-up (~ 1 year). Subjects were asked to participate in 2 sessions on 2 different days. During session 1, they sampled and rated food stimuli as described below without ingestion. During session 2, they were offered a preload of Mac & Cheese and a pudding to consume *ad libitum*. The maximum interval between the two sessions was 7 days. We successfully implemented a similar experimental design in our previous study [17].

**Session 1.** Subjects presented to the laboratory between 9 am and 3 pm. They were asked to arrive neither hungry nor full and to rate their hunger level upon arrival using a visual analogue scale (VAS: 0 = "I am not hungry at all" and 100 = "I have never been more hungry"). If they rated hunger greater than 30, they were given a small snack and were asked to wait 30 minutes, after which the hunger ratings were repeated. We felt it important to test subjects in the absence of hunger or satiety in order to minimize homeostatic effects on food liking. Before testing, each subject was trained to use the general Labeled Magnitude Scale (gLMS) to rate overall intensity, and sweetness [46], the Labeled Hedonic Scale to rate liking or disliking (LHS) [47], and the VAS to rate hunger, fullness, thirst, oiliness, fattiness, creaminess, and wanting of the stimuli. The gLMS is a computerized psychophysical tool that requires subjects to rate the perceived intensity of a stimulus along a vertical axis lined with adjectives that are spaced semi-logarithmically, based upon experimentally determined intervals to yield ratio-quality data. The LHS was derived using similar methods as the gLMS but asks subjects to rate hedonic liking or disliking [47].

The pudding and Jell-O stimuli were presented in 3 blocks with the order of presentation randomized (i.e., 3 blocks of Jell-O followed by 3 blocks of pudding). Subjects sampled 5 cc of the Jell-O at the tip of a spoon and expectorated without swallowing; likewise, they sampled around 3–5 cc of the pudding at the tip of a spoon without swallowing. After tasting each sample, subjects used the scales to rate their perceptions. They rinsed in between samples and paused for 30 seconds before tasting the next sample. Subjects were blindfolded during the experiment to prevent visual stimulation (i.e., color and consistency of the stimuli) from biasing their ratings. At the end of the session, another set of hunger, fullness, and thirst ratings were obtained.

**Session 2.** Subjects presented hungry around lunchtime between 12 and 2 pm. They were asked to eat breakfast and then refrain from eating anything until the time of testing. They were allowed to drink unflavored water up until an hour before the session. First, percent body fat was assessed using air displacement plethysmography (Bod-Pod, COSMED, Italy). Since the percent body fat considered healthy differs in men and women (21–25% range in men and 30–35% range in women) [48,49], we normalized (i.e., divided) the absolute output values from plethysmography by 31% for women and 21% for men. Immediately after, subjects rated

hunger, fullness, and thirst, and were offered a set amount of Mac&Cheese as a caloric preload. The preload was intended to mimic a meal and its caloric content was at 25% of their basal metabolic rate. Next, they were offered the pudding to which they gave the highest liking rating during session 1 and instructed to eat as much as they liked. Hunger, fullness and thirst were rated following the *ad libitum* pudding consumption. Subjects also provided ratings for intensity, liking, sweetness, familiarity, oiliness, fattiness, creaminess, saltiness and wanting after consumption.

**Questionnaires.** Subjects were asked to fill out feeding behavior questionnaires, and patients also completed pain related questionnaires. Back-pain was assessed using the following pain questionnaires and scales: VAS for pain intensity, the short-form of the McGill Pain Questionnaire (MPQ) [50], the Washington Neuropathic Pain Scale (NPS) [51], and the pain DETECT questionnaire [52]. All subjects filled out the Beck's Anxiety Index (BAI) [53], the Beck's Depression Index (BDI) [54], and the Edinburgh Inventory Handedness Scale (EIHS). Feeding behavior was assessed in all subjects using the following questionnaires: impulsivity was measured with the Barratt Impulsiveness Scale Version 11 (BIS-11) [55]; eating style was measured with the Dutch Eating Behavior Questionnaire (DEBQ) [56], the Three-Factor Eating Questionnaire (TFEQ) [57], the Power of Food Scale (PFS)[58], and the Binge Eating Scale (BES) [59]; reward sensitivity was measured with Behavioral Inhibition System/Behavioral Activation System (BIS/BAS) [60]; hours of physical exercise in the past 7 days was reported using the International Physical Activity Questionnaire (IPAQ) [61]; specific food addiction problems were assessed by using Yale Food Addiction Scale (YFAS) [62].

## MRI data acquisition parameters

Participants underwent an anatomical T1-weighted scan. Siemens 3.0 T Trio magnet equipped with a 32-channel head-coil was used to acquire the images. MPRAGE 3D T1-weighted acquisition sequence was as follows: TR/TE = 1900/2.52 ms, flip angle, 9˚, matrix 256 x 256 with 176-1mm slices.

## Calculations of nucleus accumbens volume

Structural data were analyzed with the standard automated processing stream of the Functional Magnetic Resonance Imaging of the Brain (FMRIB) software library (FSL) 5.0.10, which shows high reliability across laboratories [63]. The analysis sequence includes skull extraction, a two-stage linear subcortical registration, and segmentation using Integrated Registration and Segmentation Tool (FIRST) [64], which is part of FMRIB. The volumes of right and left nucleus NAc were calculated for each participant and normalized to standard Montreal Neurologic Institute (MNI) space. The normalization coefficient was calculated using FSL SIENAX [65].

**Statistical analysis.** Statistical analysis was performed with STATISTICA 13 (Stat Soft, Inc.). We performed between-group analyses using unpaired t-tests and 2-way mixed measures ANOVAs with stimulus concentration considered as a repeated within subject measure. Ratings were averaged across the 3 presentations for each stimulus concentration. Paired t-tests were used to examine within-group effects (e.g. hunger before and after each session) and regression analyses to investigate the relationships between caloric intake or volume and internal state or psychophysical ratings. Correction for sex is indicated when included. Identification and exclusion of outliers followed the Tukey method [66]. Figs were generated using Sigmaplot 14.0. Bonferroni was used for repeated measurement corrections. Since our starting hypothesis centered around disrupted hedonic ratings in low-back pain patients, differences in liking ratings were not corrected for multiple comparisons and were considered significantly different if $p < 0.05$. Comparisons for other ratings were corrected for multiple comparisons

when reported as significant (i.e., p < 0.05/8 = 0.00625, where 8 is the number of different food attributes we collected). All relevant data are within the manuscript and its S1–S7 Figs and S1–S10 Tables.

## Results

### Demographic and clinical characteristics

Demographic and clinical data for all participants at both baseline and follow-up are presented in **Tables 1–3**. At baseline, the average age, sex distribution, handedness, BMI, percentage

**Table 1. Demographic and clinical characteristics of all groups baseline [a].**

| Parameter | SBP (n = 51) | CLBP (n = 43) | HC (n = 36) | P Value [b] |
|---|---|---|---|---|
| Age, y | 31.20 ± 1.65 | 33.79 ± 1.81 | 31.31 ± 1.97 | 0.46 |
| Male sex n, (%) | 31 (60.8%) | 17 (39.5%) | 20 (55.6%) | 0.11 |
| EHIS | 75.12 ± 6.74 | 74.03 ± 9.06 | 73.94 ± 7.52 | 0.45 |
| BMI, kg/m$^2$ | 26.16 ± 0.80 | 24.38 ± 0.75 | 24.88 ± 0.86 | 0.38 |
| % Body fat | 1.05 ± 0.06 | 0.95 ± 0.07 | 1.03 ± 0.07 | 0.56 |
| Years of education | 15.65 ± 0.55 | 15.32 ± 0.63 | 15.16 ± 0.70 | 0.85 |
| BDI | 5.27 ± 1.67 | 7.61 ± 1.45 | 2.66 ± 1.98 | **0.003**[*] |
| BAI | 6.63 ± 1.92 | 7.91 ± 1.95 | 3.41 ± 2.29 | 0.07 |
| BIS-11 | 55.61 ± 1.83 | 56.00 ± 2.29 | 55.48 ± 2.18 | 0.98 |
| BIS | 2.52 ± 0.12 | 2.52 ± 0.13 | 2.20 ± 0.10 | 0.05 |
| BAS | 1.93 ± 0.13 | 2.27 ± 0.16 | 2.24 ± 0.15 | 0.16 |
| TFEQ—Restraint | 9.24 ± 0.77 | 10.46 ± 0.95 | 8.57 ± 0.89 | 0.33 |
| TFEQ—Disinhibition | 5.15 ± 0.48 | 5.52 ± 0.57 | 4.53 ± 0.56 | 0.45 |
| TFEQ—Hunger | 5.22 ± 0.54 | 4.86 ± 0.65 | 4.17 ± 0.63 | 0.44 |
| PFS—Food availability | 11.63 ± 0.94 | 11.68 ± 1.19 | 12.52 ± 1.12 | 0.81 |
| PFS—Food presence | 14.32 ± 0.95 | 13.42 ± 1.15 | 15.86 ± 1.13 | 0.30 |
| PFS—Food tasted | 13.12 ± 0.76 | 12.45 ± 0.89 | 13.48 ± 0.90 | 0.70 |
| BES | 4.98 ± 2.31 | 11.72 ± 5.43 | 5.04 ± 2.76 | 0.15 |
| DEBQ | 1.99 ± 0.14 | 2.08 ± 0.15 | 1.85 ± 0.17 | 0.62 |
| IPAQ, h | 165.71 ± 24.22 | 156.2 ± 23.39 | 157.07 ± 28.08 | 0.96 |
| YFAS | 1.15 ± 0.12 | 1.55 ± 0.20 | 1.23 ± 0.17 | 0.22 |
| VAS pain | 3.38 ± 0.29 | 4.50 ± 0.33 | - | 0.010[*] |
| Pain duration, weeks | 9.82 ± 29.10 | 281.9 ± 45.43 | - | <0.0001 |
| Pain DETECT questionnaire | 6.18 ± 0.86 | 7.42 ± 0.96 | - | 0.28 |
| MPQ—Total | 8.81 ± 0.84 | 10.11 ± 0.76 | - | 0.26 |
| MPQ—Sensory | 6.47 ± 0.72 | 7.30 ± 0.69 | - | 0.42 |
| MPQ—Affective | 2.43 ± 0.48 | 2.80 ± 0.55 | - | 0.56 |
| NPS—Total | 21.82 ± 2.09 | 28.20 ± 2.43 | - | 0.045[*] |

*Abbreviations*: CLBP, chronic low back pain; SBP, subacute back pain; HC, Healthy Control; EHIS, Edinburgh Handedness Inventory Score; BMI, body mass index; BDI, Beck Depression Index; BAI, Beck Anxiety Index; BIS-11, Barratt Impulsiveness Scale; BIS, Behavioral Inhibition System; BAS, Behavioral Activation System; TFEQ, Three Factor Eating Questionnaire; PFS, Power of Food Scale; BES, Binge Eating Scale; DEBQ, Dutch Eating Behavioral Questionnaire; IPAQ, International Physical Activity Questionnaire; VAS, visual analog scale; MPQ, McGill Pain Questionnaire; PFS, Power of Food Scale; NPS, Neuropathic Pain Scale; YFAS, Yale Food Addiction Scale.

**a**, Values are expressed as mean ± standard error of mean or n (%).

**b**, Calculated by t test, Mann-Whitney Rank Sum Test or $\chi^2$ test.

Note:

[*] Statistically significant; data available for years of education: 32 in SBP, 24 in CLBP, and 19 in HC; Subjects who finished session 2 were included in our main results.

**Table 2. Demographic and clinical characteristics of SBP patients at baseline [a].**

| Parameter | SBPr (n = 20) | SBPp (n = 16) | HC (n = 36) | P Value [b] |
|---|---|---|---|---|
| Age, y | 30.60 ± 2.20 | 34.30 ± 2.40 | 30.60 ± 2.00 | 0.35 |
| Male sex n, (%) | 12 (60.0%) | 10 (62.5%) | 20 (55.6%) | 0.88 |
| EHIS | 76.75 ± 9.28 | 70.00 ± 10.37 | 62.67 ± 7.88 | 0.50 |
| BMI, kg/m$^2$ | 24.75 ± 1.15 | 26.79 ± 1.28 | 25.36 ± 0.93 | 0.48 |
| % Body fat | 0.94 ± 0.09 | 1.13 ± 0.10 | 1.03 ± 0.07 | 0.36 |
| Years of education | 15.71 ± 0.87 | 15.71 ± 4.53 | 15.16 ± 0.7 | 0.85 |
| BDI | 6.85 ± 1.06 | 2.81 ± 0.82 | 2.66 ± 1.98 | **<0.001***|
| BAI | 7.80 ± 1.84 | 4.50 ± 0.74 | 3.41 ± 2.29 | **0.04*** |
| BIS-11 | 55.50 ± 1.98 | 57.31 ± 3.19 | 55.48 ± 2.18 | 0.83 |
| BIS | 2.13 ± 0.17 | 2.30 ± 0.12 | 2.20 ± 0.10 | 0.23 |
| BAS | 2.00 ± 0.20 | 1.88 ± 0.12 | 2.24 ± 0.15 | 0.45 |
| TFEQ—Restraint | 8.85 ± 1.09 | 8.81 ± 1.17 | 8.57 ± 0.89 | 0.93 |
| TFEQ—Disinhibition | 5.15 ± 0.59 | 5.50 ± 0.87 | 4.53 ± 0.56 | 0.85 |
| TFEQ—Hunger | 5.90 ± 0.84 | 4.69 ± 0.93 | 4.17 ± 0.63 | 0.27 |
| PFS—Food availability | 12.25 ± 1.11 | 11.13 ± 1.45 | 12.52 ± 1.12 | 0.83 |
| PFS—Food presence | 16.30 ± 1.32 | 12.88 ± 1.50 | 15.86 ± 1.13 | 0.24 |
| PFS—Food tasted | 14.05 ± 1.05 | 12.69 ± 1.07 | 13.48 ± 0.90 | 0.77 |
| BES | 4.79 ± 0.85 | 5.00 ± 1.26 | 5.04 ± 2.76 | 0.91 |
| DEBQ | 1.94 ± 0.23 | 2.11 ± 0.25 | 1.85 ± 0.17 | 0.81 |
| IPAQ, h | 175.85 ± 25.11 | 98.95 ± 16.47 | 157.07 ± 28.08 | 0.16 |
| YFAS | 1.30 ± 0.20 | 1.06 ± 0.22 | 1.23 ± 0.17 | 0.71 |
| VAS pain | 3.64 ± 0.40 | 3.03 ± 0.43 | - | 0.31 |
| Pain duration, week | 9.16 ± 0.94 | 9.98 ± 0.99 | - | 0.55 |
| Pain DETECT questionnaire | 6.66 ± 1.04 | 5.87 ± 1.10 | - | 0.61 |
| MPQ—Total | 9.39 ± 1.12 | 7.95 ± 1.18 | - | 0.38 |
| MPQ—Sensory | 5.83 ± 1.00 | 6.91 ± 1.06 | - | 0.47 |
| MPQ—Affective | 3.56 ± 0.67 | 1.27 ± 0.71 | - | **0.03*** |
| NPS—Total | 22.96 ± 2.78 | 20.85 ± 2.95 | - | 0.61 |

*Abbreviations*: SBPr, subacute back pain recovered; SBPp, subacute back pain persistent; HC, Healthy Control.

**a**, Values are expressed as mean ± standard error of mean or n (%).

**b**, Calculated by t test, Mann-Whitney Rank Sum Test, ANOVA, Kruskal-Wallis One Way Analysis of Variance or $\chi^2$ test.

Note:

* Statistically significant; data available for years of education: 14 in SBPr, 14 in SBPp, and 19 in HC; Subjects who finished session 2 were included in our main results.

body fat, years of education, and anxiety levels did not differ between SBP, CLBP, and HC (**Table 1**). On average, depression scores were different between the groups (BDI, p = 0.003; ANOVA), with CLBP reporting the highest score (7.61 ± 1.45). However, all three groups had less than minimal depression (BDI score < 9). There was no difference in the eating style or in the average number of hours spent in physical activity during the past 7 days before testing (IPAQ, p = 0.96) between the groups (**Table 1**). CLBP patients reported significantly higher back-pain intensity on the visual analogue scale (VAS) compared to SBP patients (CLBP = 4.50 ± 0.33; SBP = 3.38 ± 0.29, p = 0.01). SBPr and SBPp patients did not differ from HC on demographic characteristics, eating style, or physical activity at baseline or at follow-up (**Tables 2 and 3**). SBPr and SBPp patients exhibited different pain qualities. SBPp patients showed lower affective pain (SF-MPQ-Affective; SBPr = 3.6 ± 0.7 versus SBPp = 1.3 ± 0.7, p = 0.03) at baseline but higher sensory pain at follow-up (MPQ-Sensory; SBPr = 2.4 ± 0.5

**Table 3. Demographic and clinical characteristics of SBP patients at follow-up [a].**

| Parameter | SBPr (n = 16) | SBPp (n = 13) | HC (n = 16) | P Value [b] |
|---|---|---|---|---|
| Age, y | 33.69 ± 2.50 | 38.77 ± 2.77 | 31.88 ± 2.50 | 0.18 |
| Male sex n, (%) | 8 (50.0%) | 8 (61.5%) | 7 (43.75%) | 0.82 |
| BMI, kg/m$^2$ | 25.35 ± 1.00 | 27.50 ± 8.70 | 24.83 ± 1.51 | 0.46 |
| % Body fat | 0.94 ± 0.09 | 1.16 ± 0.12 | 0.96 ± 0.10 | 0.28 |
| BDI | 4.20 ± 1.34 | 3.69 ± 1.44 | 2.38 ± 3.84 | 0.60 |
| BAI | 4.67 ± 1.53 | 4.23 ± 1.64 | 3.81 ± 1.48 | 0.92 |
| BIS-11 | 52.60 ± 1.81 | 55.23 ± 2.89 | 51.56 ± 2.00 | 0.46 |
| BIS | 3.18 ± 0.40 | 2.51 ± 0.15 | 2.78 ± 0.26 | 0.22 |
| BAS | 3.23 ± 0.10 | 3.14 ± 0.13 | 3.41 ± 0.12 | 0.33 |
| TFEQ—Restraint | 9.93 ± 1.39 | 9.42 ± 1.28 | 8.94 ± 1.24 | 0.86 |
| TFEQ—Disinhibition | 4.67 ± 0.64 | 5.17 ± 1.17 | 5.00 ± 0.92 | 0.94 |
| TFEQ—Hunger | 4.20 ± 0.87 | 3.77 ± 0.89 | 4.63 ± 0.82 | 0.78 |
| PFS—Food availability | 10.60 ± 0.95 | 9.69 ± 1.21 | 11.31 ± 1.29 | 0.70 |
| PFS—Food presence | 13.33 ± 1.47 | 12.31 ± 1.46 | 12.69 ± 1.47 | 0.90 |
| PFS—Food tasted | 11.73 ± 1.08 | 11.69 ± 1.15 | 11.38 ± 1.11 | 0.97 |
| BES | 2.67 ± 1.14 | 3.00 ± 1.37 | 1.27 ± 1.07 | 0.50 |
| DEBQ | 1.67 ± 0.19 | 2.22 ± 0.25 | 1.95 ± 0.28 | 0.44 |
| IPAQ, h | 131.04 ± 13.94 | 158.40 ± 52.74 | 103.19 ± 29.51 | 0.47 |
| YFAS | 1.00 ± 0.17 | 1.00 ± 0.19 | 1.13 ± 0.18 | 0.83 |
| VAS pain | 1.45 ± 0.31 | 3.55 ± 0.47 | - | **<0.001**[*] |
| Pain duration, week | - | 93.62 ± 11.23 | - | - |
| Pain DETECT questionnaire | 4.01 ± 0.90 | 6.74 ± 1.40 | - | 0.15 |
| MPQ—Total | 4.53 ± 1.35 | 7.99 ± 0.98 | - | 0.06 |
| MPQ—Sensory | 2.36 ± 0.54 | 5.67 ± 1.00 | - | **0.005**[*] |
| MPQ—Affective | 2.16 ± 1.09 | 2.34 ± 0.67 | - | 0.90 |
| NPS—Total | 10.90 ± 3.86 | 20.72 ± 3.52 | - | 0.07 |

**a**, Values are expressed as mean ± standard error of mean or n (%).

**b**, Calculated by t test, Mann-Whitney Rank Sum Test, ANOVA, Kruskal-Wallis One Way Analysis of Variance or $\chi^2$ test.

Note:

[*] Subjects who finished session 2 were included in our main results.

versus SBPp = 5.7 ± 1.0, p = 0.005). A significantly higher proportion of CLBP patients than SBP patients or HC reported that high-fat and/or high-sugar food are "problematic" for them on the YFAS (**S1 Fig**). Surprisingly, when we separated the SBP group into SBPp and SBPr patients at baseline and follow-up, we observed that a higher proportion of SBPr patients reported high-fat and/or high-sugar food to be problematic for them than of the HC subjects (**S2** and **S3** Figs).

## Internal state and perception of stimuli in low-back pain and healthy subjects at session 1

**Internal state.** At baseline, SBP, CLBP and HC subjects did not differ on hunger, fullness or thirst ratings (repeated measures ANOVA, group effects). Only hunger ratings showed time effects (i.e., changes of internal state before compared to after session 1) ($F_{1,105}$ = 8.00; p = 0.006), and there was no group by time interaction for hunger, fullness, or thirst ratings between the groups. (**S4 Fig** left and **S1 Table**). SBPr, SBPp, and HC groups did not differ on hunger, fullness, and thirst ratings during session 1 at baseline. There was neither a time effect

nor a group by time effect (**Fig 2A–2C left, S2 Table**). At follow-up, during session 1, SBPr, SBPp and HC did not differ on hunger, fullness, or thirst ratings. There was no significant time effect or group by time effect except for hunger ratings which showed a significant time effect ($F_{1,40}$ = 8.27, p = 0.006) (**Fig 2A–2C right, S3 Table**).

**Pudding ratings.** Puddings containing 4 different concentrations of fat (0%, 3.1%, 6.9% and 15.6% weight by weight) were sampled without ingestion during session 1. At baseline, SBP, CLBP and HC subjects exhibited no difference in pudding liking (**Fig 3A left**), intensity (**S5A Fig left**), sweetness, familiarity, fattiness, creaminess, oiliness, or wanting of pudding (mixed-measures ANOVA, group effect, **S4 Table**). However, the three groups showed a group x concentration interaction (p = 0.042) for ratings of pudding liking, and concentration effects for ratings of liking (p < 0.005) (**Fig 3A left, S4 Table**). Similarly, SBPr, SBPp and HC subjects did not differ on any of the pudding ratings at baseline (**Fig 3B left, S5 Table**), but the 3 groups showed a group x concentration interaction (p = 0.005) for ratings of pudding liking driven mostly by a decrease in liking ratings reported by SBPr patients for increasing concentrations of fat (**Fig 3B left, S5 Table**). In addition, creaminess increased significantly with fat concentration (p < 0.001) (**S5 Table**). At follow-up, SBPr, SBPp an HC groups differed on pudding liking ratings (p < 0.05), a difference driven mostly by the lower liking ratings of SBPr patients (**Fig 3C left, S6 Table**). Here, also, creaminess increased significantly with fat concentration (p < 0.005) (**S6 Table**).

**Jello ratings.** At baseline, SBP, CLBP and HC subjects did not differ on liking (**Fig 3A middle**) or any other rating of flavored jello (**S5A Fig middle and S4 Table**). These results are consistent with our previous findings of normal ratings of sweetened food in CLBP patients [17]. There was a strong increase in ratings of jello attributes with increasing concentration of sucrose, except for oiliness, but no significant group x concentration effect (**Fig 3A middle, S5A Fig and S4 Table**). Similarly, SBPr, SBPp and HC groups did not differ on any of the flavored Jell-O ratings but showed a strong increase in the ratings of Jell-O attributes with increasing concentration of sucrose at both baseline and follow-up (**Fig 3B and 3C middle, S5B and S5C Fig and S5 and S6 Tables**)

## Low-back pain disrupts the association between subjective liking and caloric consumption

On the day of the ad libitum feeding assessment (session 2), we collected our subjects' internal state ratings (hunger, fullness, and thirst) before and after consumption and ratings of the food attributes after consumption.

**Internal state.** At baseline, SBP, CLBP and healthy subjects did not differ on hunger, fullness, or thirst (**S7 Table**). Ratings of hunger and fullness but not thirst changed (p < 0.001) after a preload of Mac & Cheese and *ad libitum* pudding consumption (time effect) in session 2 (**S4A–S4C Fig right and S7 Table**). Internal state ratings did not differ at baseline or follow-up between SBPr, SBPp and HC subjects (**Fig 4A–4C, S7 Table**). Hunger and fullness but not thirst ratings changed significantly (p < 0.001) following Mac & Cheese and *ad libitum* pudding consumption (time effect) at both baseline and follow-up (**S7 Table, Fig 4A–4C middle and right**).

**Foods ratings.** Ratings of liking, intensity, sweetness, familiarity, saltiness, fattiness, creaminess, oiliness, or wanting for Mac&Cheese and pudding did not differ among CLBP, SBP and HC groups (**S8 Table and S6 Fig**). These ratings were also not different between SBPp, SBPr and HC groups at baseline or follow-up (**S9 and S10 Tables and S6 Fig**). CLBP, SBP and HC groups' total caloric consumption from preferred puddings was not significantly different (p = 0.33) (**Table 4**). We wanted to investigate whether the magnitude of liking

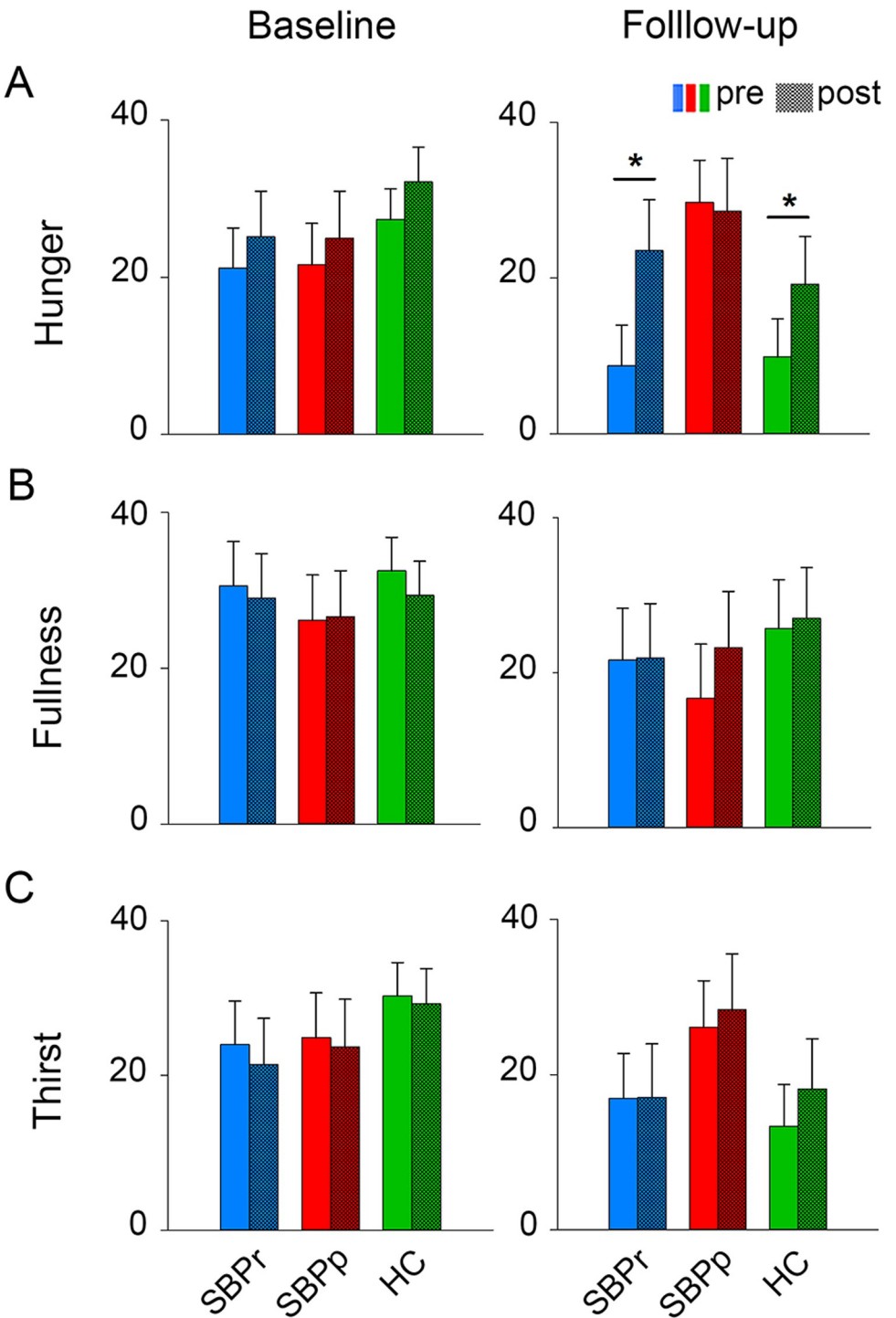

**Fig 2. Internal state ratings before and after session 1.** (**A**) Hunger, (**B**) Fullness, and (**C**) Thirst rated by SBPr, SBPp and HC (baseline, left; follow-up, right). There were no time and groups effects in any of the other analyses. *Abbreviations*: *SBPr, recovered subacute back pain patients; SBPp, persistent subacute back pain patients; HC, Healthy Control.* *, p < 0.05.

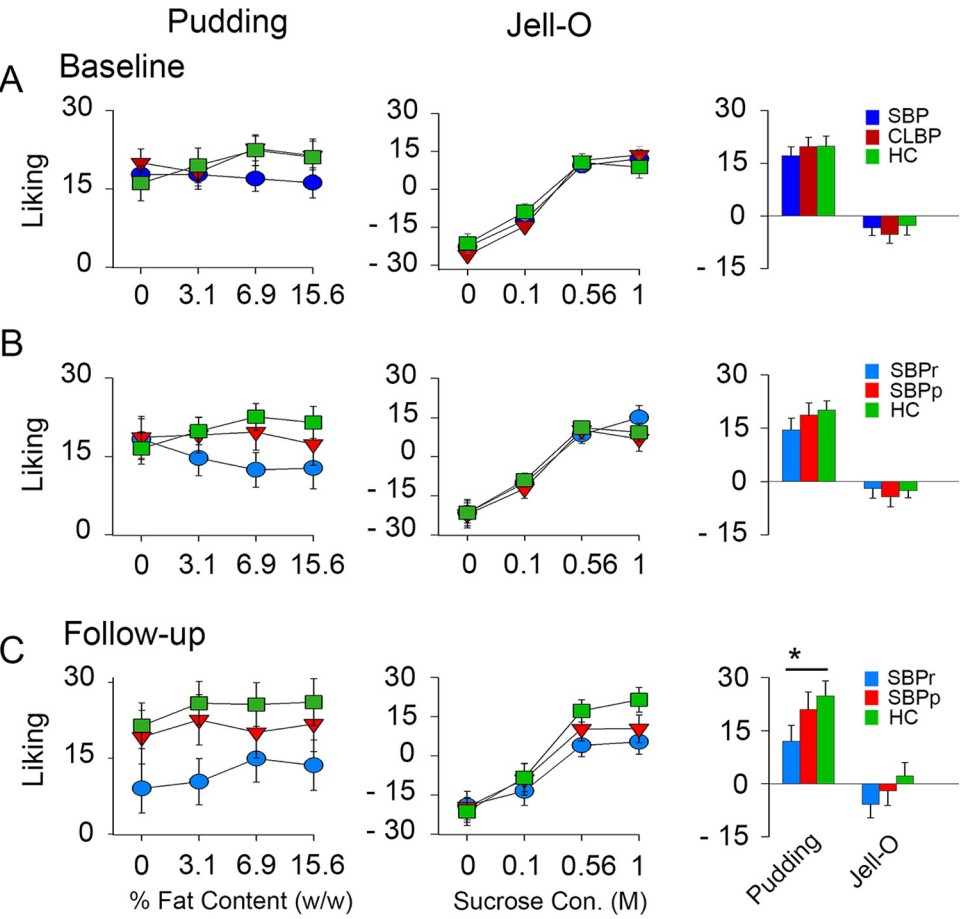

**Fig 3. Psychophysical ratings (liking) during session 1 at baseline and follow-up.** (**A**) shows results for SBP (n = 51), CLBP (n = 43 for pudding and n = 41 for Jello) and healthy control subjects (n = 36) at baseline; (**B**) shows results for SBPr (n = 20), SBPp (n = 16) and HC (n = 36) at baseline; and (**C**) for SBPr (n = 15 for pudding and n = 13 for Jello), SBPp (n = 13) and HC (n = 16 for pudding and n = 14 for Jello) at follow-up. (**A–C**) (**Left**) Participants' ratings for each of the 4 fat concentrations of puddings. (**Middle**) Ratings for each of the 4 sucrose concentrations of flavored Jello. (Far right) Bar plot depicting mean ± standard error of the mean (SEM) across all the concentrations for pudding (left) and jello (**Right**). No significant difference were found among groups. Adding covariates, including pretest hunger rating and sex, did not alter the main results.

ratings of the preferred pudding collected during session 1 predicted caloric intake when participants came back for pudding consumption during session 2 as we previously studied [17]. Liking of preferred pudding predicted significantly caloric intake in HC (r = 0.60, p = .001) but not in SBP (r = 0.14, p = 0.40) or CLBP patients (r = 0.13, p = 0.49), indicating, as hypothesized, that the relationship between liking of food and caloric ingestion is absent in the patients' groups (**Fig 5A**).

SBPr, SBPp and HC groups' total caloric consumption from preferred puddings did not significantly differ at baseline (p = 0.44), or follow-up (p = 0.22) (**Table 4**). Interestingly, at baseline the prediction of caloric intake from the magnitude of liking ratings of the most preferred pudding sampled during session 1 was close to significance and stronger in the at risk SBPp patients (r = 0.38, p = 0.16, **Fig 5B right**) than in SBPr patients (r = - 0.14, p = 0.53, **Fig 5B middle**). This picture persisted at follow-up. As such, the prediction of caloric intake from the magnitude of liking ratings was close to significance in SBPp patients, (r = 0.52, p = 0.08, **Fig 5C right**) but was weaker and was not statistically significant in SBPr patients (r = 0.21,

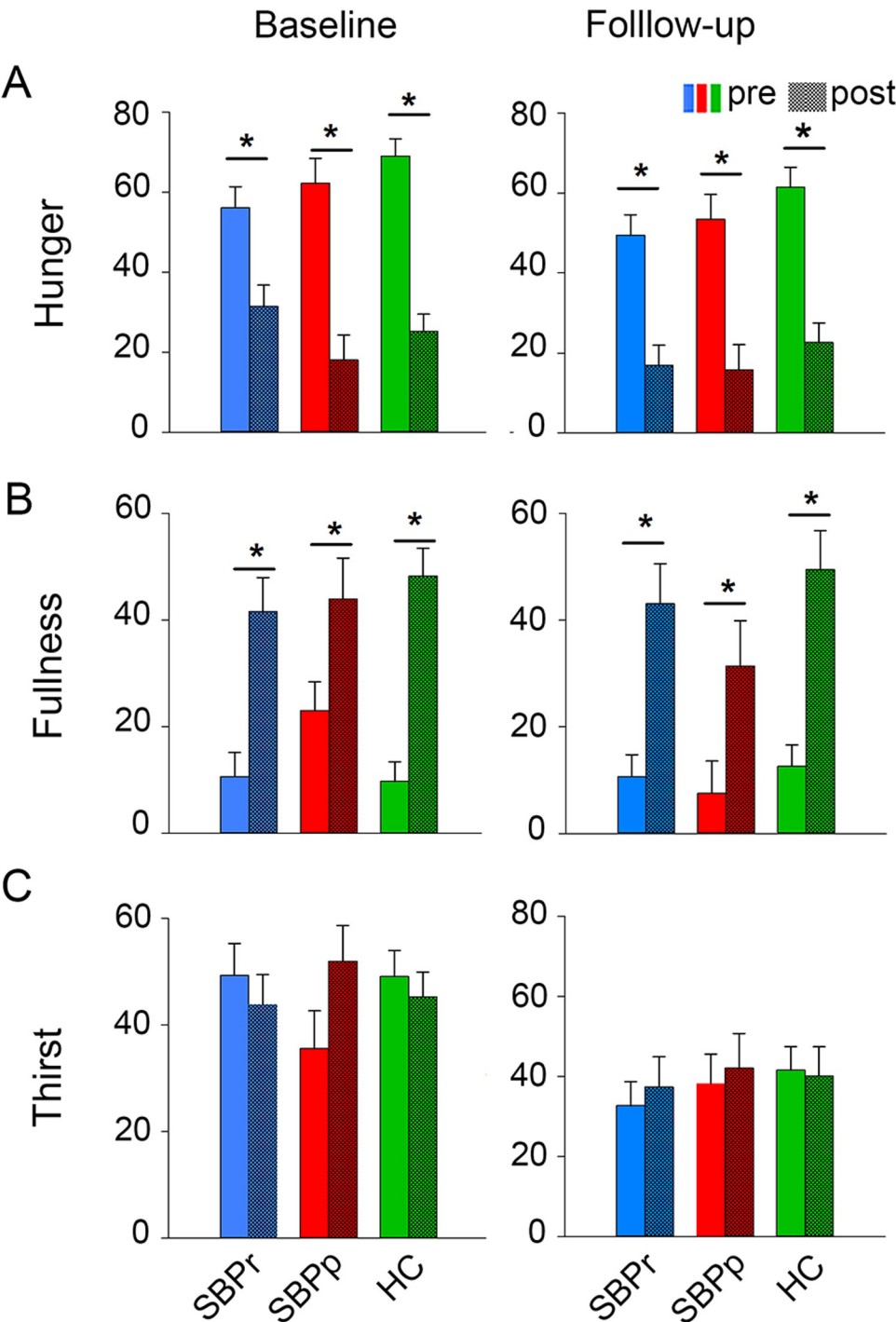

**Fig 4. Internal state ratings before and after the *ad libitum* consumption of the preferred pudding during session 2.** (**A**) Hunger, (**B**) Fullness, and (**C**) Thirst rated by SBPr, SBPp and HC (baseline, left; follow-up, right). There were no time or group effects in any of the analyses. Hunger and fullness ratings changed significantly after pudding consumption in all groups (**A-B**); thirst ratings did not differ following between groups or after consumption. There was no group or interaction effect during session 2.

**Table 4. Caloric consumption during session 2.**

| | Baseline | | Follow-up | |
|---|---|---|---|---|
| | **Calories from Mac and Cheese** | **Calories from Pudding** | **Calories from Mac and Cheese** | **Calories from Pudding** |
| CLBP | 367.18 ± 16.25 | 207.92 ± 40.39 | - | - |
| SBP | 359.62 ± 18.16 | 271.74 ± 44.50 | | |
| HC | 388.06 ± 18.53 | 296.00 ± 46.63 | | |
| *Group Effect* | *0.51* | *0.33* | *-* | *-* |
| SBPr | 363.65 ± 22.36 | 237.30 ± 60.80 | 370.75 ± 26.96 | 252.15 ± 77.13 |
| SBPp | 372.33 ± 26.73 | 185.72 ± 72.66 | 333.37 ± 29.81 | 190.59 ± 61.44 |
| HC | 388.06 ± 18.53 | 296.00 ± 46.63 | 386.95 ± 25.65 | 358.97 ± 64.33 |
| *Group Effect* | *0.69* | *0.44* | *0.39* | *0.22* |

p = 0.45, **Fig 5C, middle**). The same relationship remained significant in healthy subjects at follow-up (r = 0.56, p = 0.025, **Fig 5C left**). It is worth noting that the correlation between liking and caloric intake in SBPr patients showed a trend of recovery between baseline and follow-up becoming stronger in magnitude and changing from negative to positive, and hence

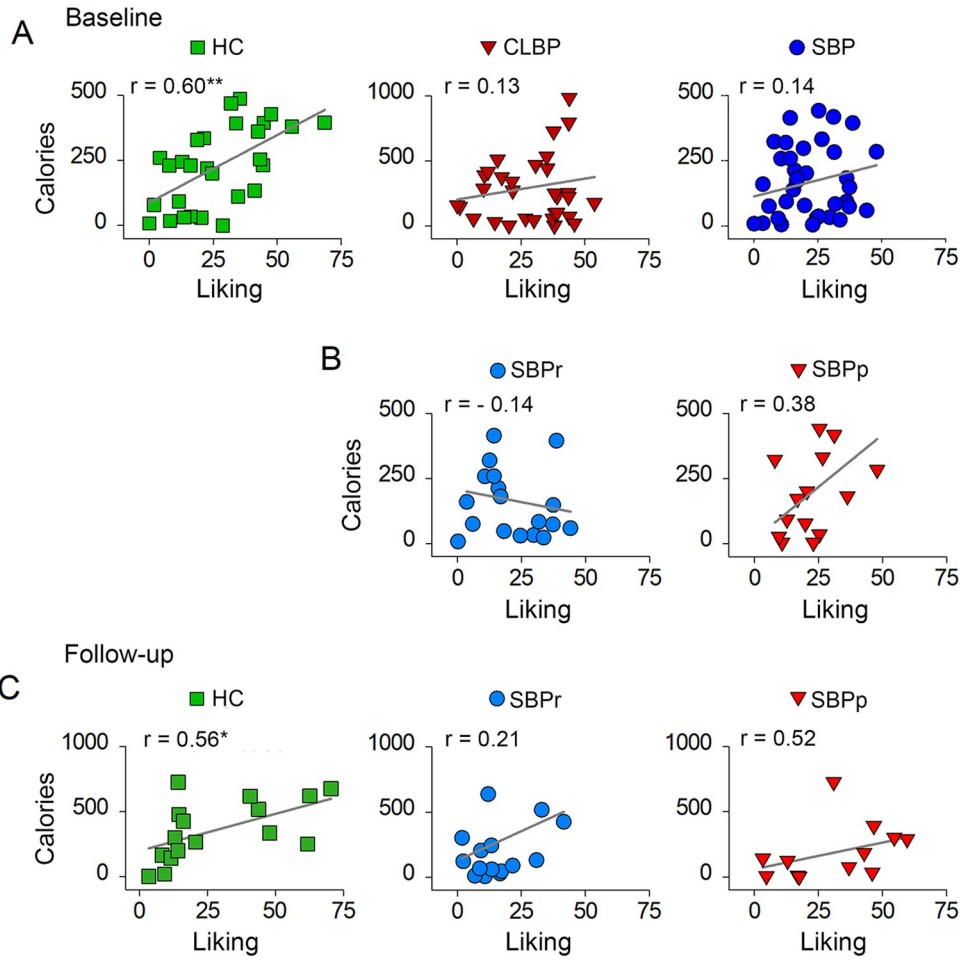

**Fig 5.** Regression plots of the most preferred pudding liking reported during session 1 and the caloric ingestion from that pudding during session 2 at baseline (A-B) and at follow-up (C). (A) Regression plots for SBP (n = 37), CLBP (n = 32) and healthy control subjects (n = 28) at baseline; (**B**) Regression plots for SBPr (n = 17) and SBPp (n = 14) groups at baseline; (**C**) Regression plots for SBPr (n = 15), SBPp (n = 12) and HC (n = 15) subjects at follow-up. **\*\***, p < 0.01, \*, p < 0.05.

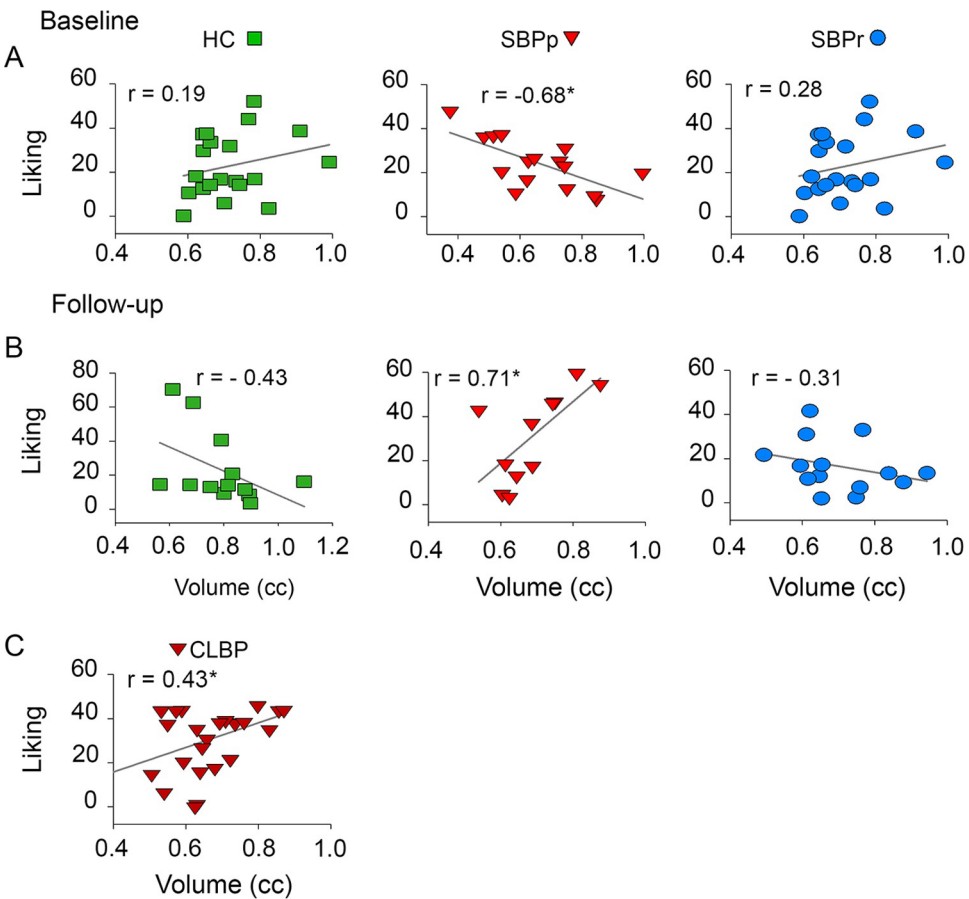

**Fig 6. Nucleus accumbens volume in centimeter cubes (cc) predicts highly palatable food liking only in patients at risk for chronic pain (SBPp) and in CLBP patients.** (**A-C**) Regression (Spearman rank) between LNAc volume and average liking derived from the most liked pudding sampled in session 1 at baseline and in (**D-F**) at follow-up. (**G**) Regression between LNAc volume and liking in CLBP patients. *, p < 0.05.

becoming more similar to the one observed in HC (**Fig 5B and 5C**, **middle**). Finally, we tested for any significant change in BMI or body fat content in SBPp, SBPr, and HC between baseline and follow-up. There were no significant changes or group x time interaction (**S7 Fig**).

### Hedonic perception of high fat pudding is associated with the limbic brain in SBPp and CLBP patients only

The LNAc volume was significantly and strongly correlated with the magnitude of liking ratings of high fat pudding in SBPp and CLBP patients only (**Fig 6**). While the correlation was large at baseline (r = - 0.68, p < 0.05) and follow-up (r = 0.71, p < 0.05) in SBPp patients, it changed direction from negative to positive respectively and, at follow-up, was consistent in direction with the correlation observed in CLBP patients (**Fig 6**). The volume of NAc did not predict caloric intake from pudding in any of our groups.

## Discussion

This work adds evidence to our hypothesis [17] that back pain patients' eating behavior differs significantly from the eating behavior of healthy subjects not in pain, and affects specifically hedonic processing [27,29,30]. The aim of this work is to further understand whether back

pain patients diverge from healthy subjects in their eating behavior with the onset of sub-acute back-pain or as the pain becomes chronic. Surprisingly, while at baseline hedonic ratings and the relationship of food liking to caloric intake were mostly preserved in SBPp patients, the relationship of liking to caloric intake was disrupted (absent) in SBPr patients. At follow-up, hedonic blunting worsened in SBPr patients, but their satiety signals started looking more like those of healthy controls, while SBPp patient's eating behavior remained practically within normal limit. We did not observe hedonic blunting in the CLBP patients but we confirmed the absent relationship between liking of highly palatable fat-rich food and caloric consumption from that food as we have previously shown [17]. We also confirmed that hedonic blunting in SBPp and CLBP patients does not generalize to sweetened food. In addition to the behavioral findings, more CLBP patients reported on the YFAS [67] that high-fat/high carbohydrate foods (e.g. ice cream, cookies) were problematic for them than SBP patients or healthy controls. These reports are consistent with our hypothesis that altered cortico-limbic [25,26,44,68–75] pathways in CLBP patients entails altered food choices and eating behavior. Our results suggest, therefore, that eating behavior changes dynamically over the course of development of CLBP, and disruptions set in after the onset of the chronic phase since the SBPp patients show normal eating behavior up to one year after the onset of pain.

An important finding in this work is the observation that the presence of back-pain in the sub-acute phase and the early-chronic phase is not sufficient by itself to explain the abnormal eating behavior in SBP patients. While SBPr patients showed signs of blunted hedonic ratings and disrupted satiety signals (i.e., absent relationship of liking to caloric intake), SBPp patients showed normal eating behavior. Both groups reported similar average low-back pain intensity at baseline, but SBPr patients' average back-pain was only 1.5, while SBPp patients' average back-pain was 3.5 at follow-up. SBPr patients' cortico-limbic pathways have shown signs of resilience according to neuroimaging studies. For example, SBPr patients showed widespread increased fractional anisotropy in major white matter bundles even higher than in healthy controls [76]; in addition, in a previous report on the same SBPr patients studied here, we found no difference in their nucleus accumbens volume or activity compared to healthy controls and a significant increase in their amygdala volume at baseline compared to SBPp patients, [26] which did not persist at follow-up. A functional accumbens is needed in animals to associate aversive or rewarding cues with outcomes [77]. A normal accumbens in SBPr patients allows therefore low-back pain to disrupt the relationship between hedonic perception and caloric intake. A dysfunctional accumbens in SBPp patients, on the other hand, protects them from this disruption, where hedonic ratings become strongly tied to the nucleus accumbens properties as in CLBP patients. This is evidenced by the strong correlation between LNAc volume and hedonic perception in SBPp and CLBP patients only. Hence, plastic changes in cortico-limbic pathway associated with resilience to pain could initially disrupt eating behavior in SBPr patients, which then recovers with time as back-pain subsides. It is notable also that the relationship between hedonic ratings and NAc volume changes direction in all groups from baseline to follow-up. We ascribe this change to learning mechanisms where a first exposure leads to memory formation which is then re-activated at the follow-up experiment. The fact that this change in direction is across the board indicates that it is not a confounding effect.

The lack of blunting of hedonic rating in the CLBP cohort or in the SBPp patients at follow-up observed in this work is inconsistent with our previous report [17]. This discrepancy may be explained by important changes in the experimental testing procedures; in this work, all participants were blinded while tasting food stimuli in session 1. In our previous work, we blinded participants only when they were tasting the chocolate flavored pudding and unblinded them when they were tasting vanilla flavored pudding because we were able to make all vanilla flavored stimuli look alike. It seems, therefore, that the cognitive-perceptual

visual processing of offered stimuli has a sizable effect on how patients rate their hedonic experience. This also implies that SBPr patients could show further hedonic blunting were we to shift back to the previous testing procedures.

Others and we have shown that structural and functional plasticity in the NAc is directly involved in CLBP [26,44,68,70]. As such, a smaller accumbens and an increased accumbens to medial prefrontal cortex functional connectivity are associated with a higher risk of transitioning to CLBP [26,44]; in addition, the accumbens exhibits a signature loss of low frequency fluctuations in CLBP that is not observed during the sub-acute phase [26]. On the other hand, the accumbens role in mediating hedonic reactions to food [40,78] and fat intake via μ-opioid binding is well established [79]. The blunted hedonic reactions and disrupted satiety signals affecting highly palatable fat-rich foods in back-pain patients are therefore consistent with the overlapping neurobiology of back-pain and pleasure systems in the brain. Notably, in both this work and our previous report [17], we did not observe any group differences pertaining to the sensory experience of the presented stimuli or the hedonic ratings of sweetened food, consistent with the described role of μ-opioid binding in the accumbens in selectively enhancing fat intake [80]. The normal eating behavior concerning sweetened foods is also consistent with a recent rodent study [81] showing that rats with neuropathic pain have normal hedonic and motivational responses to sweetened food (sucrose water and sucrose pellets). Future studies will need to test whether μ-opioid receptor activation or blockade alters hedonic responses in CLBP patients by, for example, using an opioid agonist like morphine or an opioid antagonist like naloxone.

Increased anhedonia predicts long-term increase in BMI [82,83] in individuals not suffering from clinical pain. We did not observe a change in BMI or adiposity in SBPr, SBPp and HC groups at follow-up. It remains to be determined, therefore, whether abnormal hedonic signals accounts for the association between obesity and CLBP long-term. Nevertheless, BMI and adiposity were matched between the groups and are, therefore, unlikely to account for the observed differences in eating behavior. Substance misuse is another condition associated with motivational disruptions, and anhedonia is associated with increased substance use. Consistent with our current and previous findings [17], Garland et al. [27], have reported that chronic pain patients exhibit anhedonia independently of opioid prescription status, but that chronic pain patients misusing opioids exhibit significantly higher anhedonia than medication free or opioid-treated patients who were not misusing their prescription. Our results are complementary to these findings and offer a novel behavioral approach to directly measure anhedonia which is usually assessed by questionnaires [84].

In summary, we uncovered a complex relationship between hedonic perception of highly palatable fat-rich food and the development of CLBP and reproduced our previous finding [17], namely that, unlike healthy controls, hedonic responses to fat-rich food fail to predict caloric intake in low-back pain patients. We also find a strong and dynamic relationship between hedonic perception and the NAc volume in at-risk SBPp patients as they transition from the sub-acute to the chronic phase of back pain. Future longitudinal studies of longer duration than ours are needed to understand the long-term impact of altered eating behavior on BMI and adiposity of CLBP patients.

## Supporting information

**S1 Fig. Percentage of participants from each group reporting "problems" with dietary items listed in YFAS.** A higher proportion of CLBP subjects reported problems with food high in sugar or fat (eg. ice cream, p = 0.04; cookies, p = 0.02; hamburger, p = 0.02; rice, and p = 0.02; Pearson Chi-square test) than HC. The proportion of participants reporting no

problem with different food items was significantly higher in HC than CLBP. The proportion of SBP patients was somewhere in the middle for several items (e.g. ice cream, chocolate). *Abbreviations: CLBP, chronic low back pain; SBP, subacute back pain; HC, Healthy Control; YFAS, Yale Food Addiction Scale.*
(DOCX)

**S2 Fig. Problematic food listed in YFAS for SBP groups and HC at baseline.** SBPr, subacute back pain recovered; SBPp, subacute back pain persistent; HC, Healthy Control; YFAS, Yale Food Addiction Scale.
(DOCX)

**S3 Fig. Problematic food listed in YFAS for SBP groups and HC at follow-up.**
(DOCX)

**S4 Fig. Internal ratings of SBP, CLBP and HC subjects during session 1 (left) and 2 (right).** (**A**) Hunger rated by SBP, CLBP and HC participants; (**B**) Fullness; (**C**) Thirst.
(DOCX)

**S5 Fig. Psychophysical ratings (intensity) during session 1 at baseline and follow-up.** A for SBP (n = 51), CLBP (n = 43 for pudding and n = 41 for Jello) and healthy control subjects (n = 36) at baseline; B for SBPr (n = 20), SBPp (n = 16) and HC (n = 36) at baseline and C for SBPr (n = 15 for pudding and n = 13 for Jello), SBPp (n = 13) and HC (n = 16 for pudding and n = 14 for Jello) at follow-up. (A–C) (Left) Participants' ratings for each of the 4 fat concentrations of puddings. (Middle) Ratings for each of the 4 sucrose concentrations of orange-flavored Jello. (Far right) Bar plot depicting mean ± standard error of the mean (SEM) across all the concentrations for pudding (left) and juice (right). No group effect, concentration effect or interaction was found. Adding covariates, including pretest hunger rating and sex did not affect the main results.
(DOCX)

**S6 Fig. Psychological ratings during session 2 at baseline and follow-up.** (**A**) Ratings of MacCheese (left) and pudding (right) for SBP, CLBP and HC participants at baseline; (**B**) Ratings of MacCheese (left) and pudding (right) for SBPr, SBPp and HC participants at baseline; (**C**) Ratings of MacCheese (left) and pudding (right) for SBPr, SBPp and HC at follow-up. CLBP, chronic low back pain; SBP, subacute back pain; SBPr, subacute back pain recovered; SBPp, subacute back pain persistent; HC, Healthy Control. No significant differences were found among groups.
(DOCX)

**S7 Fig. Body fat change from baseline to follow-up.** No changes in any of the groups were found from baseline to 1-year follow-up. SBPr, subacute back pain recovered; SBPp, subacute back pain persistent; HC, Healthy Control.
(DOCX)

**S1 Table. Internal state ratings for SBP and CLBP patients, and for healthy subjects for session 1 at baseline** [a]**.** SBP (42), CLBP (37), HC (29). a Values are expressed as mean ± SEM. b Calculated by repeated measure ANOVA.
(DOCX)

**S2 Table. Internal state ratings for SBPr and SBPp patients, and for healthy subjects for session 1 at baseline** [a]**.** a Values are expressed as mean ± SEM. b Calculated by repeated measure ANOVA.
(DOCX)

**S3 Table. Internal state ratings for SBPr and SBPp patients, and for healthy subjects for session 1 at follow-up** [a]**.** a Values are expressed as mean ± SEM. b Calculated by repeated measure ANOVA.
(DOCX)

**S4 Table. Comparison of SBP, CLBP patients' and healthy subjects' ratings of puddings and jello during session 1** [a]**.** a F Values are results of a mixed 2-way ANOVA where group (SBP vs CLBP vs healthy) was a factor and stimulus concentration the repeated measure. * $p < .05$, ** $p < .005$, *** $p < .001$.
(DOCX)

**S5 Table. Comparison of SBPr, SBPp patients' and healthy subjects' ratings of puddings and jello during session 1 baseline** [a]**.** a F Values are results of a mixed 2-way ANOVA where group (SBPr vs SBPp vs healthy) was a factor and stimulus concentration the repeated measure. * $p < .05$, ** $p < .005$, *** $p < .001$.
(DOCX)

**S6 Table. Comparison of SBPr, SBPp patients' and healthy subjects' ratings of puddings and jello during session 1 follow-up** [a]**.** a F Values are results of a mixed 2-way ANOVA where group (SBPr vs SBPp vs healthy) was a factor and stimulus concentration the repeated measure. * $p < .05$, ** $p < .005$, *** $p < .001$.
(DOCX)

**S7 Table. Internal state ratings for session 2 at baseline and follow-up** [a]**.** a Values are expressed as mean ± SEM. b Calculated by repeated measure ANOVA.
(DOCX)

**S8 Table. Consumed food attribute ratings for CLBP and SBP patients, and healthy subjects at for session 2 at baseline** [a]**.** a Values are expressed as mean ± SEM. b Results of a one-way ANOVA among groups (SBP vs CLBP vs healthy).
(DOCX)

**S9 Table. Consumed food attribute ratings for SBPr, SBPp patients and healthy subjects for session 2 at baseline** [a]**.** a Values are expressed as mean ± SEM. b Results of a one-way ANOVA among groups (SBP vs CLBP vs healthy).
(DOCX)

**S10 Table. Ratings for SBPr, SBPp and HC groups at session 2 at follow-up.** a Values are expressed as mean ± SEM. b Results of a one-way ANOVA among groups (SBP vs CLBP vs healthy).
(DOCX)

**S1 Data. Raw data table.**
(XLSX)

## Author Contributions

**Conceptualization:** Ivan De Araujo, Dana Small, Paul Geha.

**Data curation:** Gelsina Stanley, Paul Geha.

**Formal analysis:** Yezhe Lin, Ivan De Araujo, Dana Small, Paul Geha.

**Funding acquisition:** Paul Geha.

**Investigation:** Yezhe Lin, Ivan De Araujo, Gelsina Stanley, Dana Small, Paul Geha.

**Methodology:** Ivan De Araujo, Dana Small, Paul Geha.

**Project administration:** Paul Geha.

**Supervision:** Dana Small, Paul Geha.

**Visualization:** Paul Geha.

**Writing – original draft:** Yezhe Lin, Ivan De Araujo, Gelsina Stanley, Dana Small, Paul Geha.

**Writing – review & editing:** Yezhe Lin, Ivan De Araujo, Dana Small, Paul Geha.

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
