## [Decision Letter · Decision Letter 0]

1 Oct 2021

PONE-D-21-22332Chronic pain precedes disrupted eating behavior in low-back pain patientsPLOS ONE

Dear Dr. Geha,

Thank you for submitting your manuscript to PLOS ONE. After careful consideration, we feel that it has merit but does not fully meet PLOS ONE’s publication criteria as it currently stands. Therefore, we invite you to submit a revised version of the manuscript that addresses the points raised during the review process.

 Please revise the MS in accordance to the Reviewers' comments.

We look forward to receiving your revised manuscript.

Kind regards,

Naim Akhtar Khan, PhD, DSc

Academic Editor

PLOS ONE

Journal Requirements:

3. Thank you for stating the following in the Acknowledgments / Funding Section of your manuscript: 

This work was supported by funds from the National Institute on Drug Abuse (NIDA: 5K08DA037525), from the Psychiatry Department at the Yale School of Medicine, the Psychiatry Department at the University of Rochester Medical Center, and the Del Monte Neuroscience Institute. 

This work was supported by funds from the National Institute on Drug Abuse (NIDA: 5K08DA037525), from the Psychiatry Department at the Yale School of Medicine, the Psychiatry Department at the University of Rochester Medical Center, and the Del Monte Neuroscience Institute. 

This work was supported by funds from the National Institute on Drug Abuse (NIDA: 5K08DA037525), from the Psychiatry Department at the Yale School of Medicine, the Psychiatry Department at the University of Rochester Medical Center, and the Del Monte Neuroscience Institute. 

None.

Additional Editor Comments:

The MS should be revised in the light of Reviewers' comments.

Reviewers' comments:

Reviewer's Responses to Questions

**Comments to the Author**

1. Is the manuscript technically sound, and do the data support the conclusions?

Reviewer #1: Yes

Reviewer #2: Yes

2. Has the statistical analysis been performed appropriately and rigorously? 

Reviewer #1: Yes

Reviewer #2: Yes

3. Have the authors made all data underlying the findings in their manuscript fully available?

Reviewer #1: Yes

Reviewer #2: Yes

4. Is the manuscript presented in an intelligible fashion and written in standard English?

Reviewer #1: Yes

Reviewer #2: Yes

5. Review Comments to the Author

Reviewer #1: This is a well done study addressing a complex relationship between the hedonic qualities of food and chronic pain along with evaluation of the NAc. The findings are in many ways counter-intuitive and somewhat difficult to appreciate but have important implictions. The main suggestion is that the abstract should be re-written to provide the interpretation of the findings regarding the role of the NAc in disruption of hedonic quality in SBPr patients while a dysfunctional accumbens in the SBPp group protects them against disruption. It would also be helpful to provide better definition in many places in the manuscript of the term "disruption" - what exactly was being disrupted in the context of the sentence? It will also be interesting to see the outcomes of the proposed opioid studies.

Reviewer #2: Yezhe Lin and collaborators have investigated whether Chronic pain precedes disrupted eating behavior in low-back pain patients

This research is original and suggest that chronic pain precedes disrupted eating behavior in low-back pain patients and that such disruption is directly related to structural changes in nucleus accumbens. In spite of that, some aspects need to be clarified.

Globaly , The manuscript is difficult to follow and deserves to be simplified or reorganized by example :

- In Methods section, I suggest to include a schematic diagram describing the groups and procedures

- Section results should be simplified and the figures numbers must follow as far as possible their order of quotation in the manuscript

- In all the figures , bar graph shall be a contrasting colour

- Legend of figure 2 (A-C) is omitted

- Insert the groups in fig 2 and Fig 5

- A few tapping errors must be corrected

6. PLOS authors have the option to publish the peer review history of their article (what does this mean?). If published, this will include your full peer review and any attached files.

Reviewer #1: No

Reviewer #2: No

---

## [Author Response · Author response to Decision Letter 0]

6 Dec 2021

We would like to thank both reviewers for their interest in our work and their constructive feedback which helped improve the manuscript. Please find below point by point response to the reviewers comments.

Reviewer #1: This is a well done study addressing a complex relationship between the hedonic qualities of food and chronic pain along with evaluation of the NAc. The findings are in many ways counter-intuitive and somewhat difficult to appreciate but have important implications. 

Thank you!

The main suggestion is that the abstract should be re-written to provide the interpretation of the findings regarding the role of the NAc in disruption of hedonic quality in SBPr patients while a dysfunctional accumbens in the SBPp group protects them against disruption. 

As per the reviewer’s suggestion we re-wrote the abstract to provide the interpretation of the role of the NAc in the disruption of hedonic processing. We now write: 

Chronic pain is associated with anhedonia and decreased motivation. These behavioral alterations have been linked to alterations in the limbic brain and could explain the increased risk for obesity in pain patients. The mechanism of these behavioral changes and how they set in in relation to the development of chronic pain remain however poorly understood. Here we asked how eating behavior is affected in low-back pain patients before and after they transition to chronic pain, compared to patients whose pain subsides. Additionally, we assessed how the hedonic perception of fat-rich food, which is altered in chronic pain patients, relates to the properties of the nucleus accumbens in this patients’ population. We hypothesized that the accumbens will be directly implicated in the hedonic processing of fat-rich food in pain patients because of its well-established role in hedonic feeding and fat ingestion, and its emerging role in chronic pain. Accordingly, we used behavioral assays and structural brain imaging to test sub-acute back pain patients (SBP) and healthy control subjects at baseline and at approximately one-year follow-up. We also studied a sample of chronic low-back pain patients (CLBP) at one time point only. We found that SBP patients who recovered at follow-up (SBPr) and CLBP patients showed disrupted eating behaviors. In contrast, SBP patients who persisted in having pain at follow-up (SBPp) showed intact eating behavior. From a neurological standpoint, only SBPp and CLBP patients showed a strong and direct relationship between hedonic perception of fat-rich food and nucleus accumbens volume suggesting that accumbens alterations observed in SBPp patients in previous works might protect them from hedonic eating disruptions during the early course of the illness. We conclude that disrupted eating behavior specifically sets in after pain chronification and is accompanied by structural changes in the nucleus accumbens. 

It would also be helpful to provide better definition in many places in the manuscript of the term "disruption" - what exactly was being disrupted in the context of the sentence? 

We thank the reviewer for this suggestion. We tried to rephrase or explain the word disruption to be more specific all through the manuscript in the revised version. 

It will also be interesting to see the outcomes of the proposed opioid studies.

We strongly agree with the reviewer on this future step. We are in the process of applying for funding to do these experiments. 

Reviewer #2: Yezhe Lin and collaborators have investigated whether Chronic pain precedes disrupted eating behavior in low-back pain patients

This research is original and suggest that chronic pain precedes disrupted eating behavior in low-back pain patients and that such disruption is directly related to structural changes in nucleus accumbens. In spite of that, some aspects need to be clarified.

Globaly , The manuscript is difficult to follow and deserves to be simplified or reorganized by example:

- In Methods section, I suggest to include a schematic diagram describing the groups and procedures

We thank the reviewer for this excellent suggestion. We now added a new figure (Figure 1) which depicts a schematic diagram describing the groups and the procedures.

- Section results should be simplified and the figures numbers must follow as far as possible their order of quotation in the manuscript

Per the reviewer’s suggestion we have now simplified the presentation of the results. We added a table (Table 4) to report the caloric intake separately from tables 1-3. We as much as possible now quote the figures in the order of their numbers. In addition we grouped supplementary tables 7-9 into one supplementary table 7 hence reducing the number of tables in the supporting information from 12 to 10. 

- In all the figures , bar graph shall be a contrasting colour.

Per the reviewer’s suggestion we changed all the colors of all our figures to ensure an adequate contrast. 

- Legend of figure 2 (A-C) is omitted

The legend of Figure 2 (now Figure 3) were inserted. 

- Insert the groups in fig 2 and Fig 5

We inserted the groups in Figures 2-5 which are now numbered Figure 3-6 because we added a new Figure 1. 

- A few tapping errors must be corrected

Typing errors were corrected.

---

## [Editor Report · Decision Letter 1]

10 Jan 2022

PONE-D-21-22332R1Chronic pain precedes disrupted eating behavior in low-back pain patientsPLOS ONE

Dear Dr. Geha,

Thank you for submitting your manuscript to PLOS ONE. After careful consideration, we feel that it has merit but does not fully meet PLOS ONE’s publication criteria as it currently stands. Therefore, we invite you to submit a revised version of the manuscript that addresses the points raised during the review process.

We look forward to receiving your revised manuscript.

Kind regards,

Naim Akhtar Khan, PhD, DSc

Academic Editor

PLOS ONE

Additional Editor Comments:

The MS has been revised as per comments of the reviewers. However, there are some linguistic errors in the MS. For example, in the revised version of the MS, in the Abstract section, sometimes, the principle clause is in past tense but the subordinate clause is still in the present tense. Please re-read the MS carefully and submit the new version for the final acceptance.
---

## [Author Response · Author response to Decision Letter 1]

15 Jan 2022

We thank the reviewers and the editor for their critiques. We have now thoroughly corrected the linguistic errors in the manuscript.

---

## [Editor Report · Decision Letter 2]

21 Jan 2022

Chronic pain precedes disrupted eating behavior in low-back pain patients

PONE-D-21-22332R2

Dear Dr. Geha,

We’re pleased to inform you that your manuscript has been judged scientifically suitable for publication and will be formally accepted for publication once it meets all outstanding technical requirements.

Kind regards,

Naim Akhtar Khan, PhD, DSc

Academic Editor

PLOS ONE

Additional Editor Comments (optional):

The revised MS is acceptable for the publication.
---

## [Editor Report · Acceptance letter]

2 Feb 2022

PONE-D-21-22332R2 

Chronic pain precedes disrupted eating behavior in low-back pain patients 

Dear Dr. Geha:

I'm pleased to inform you that your manuscript has been deemed suitable for publication in PLOS ONE. Congratulations! Your manuscript is now with our production department. 

Kind regards, 

on behalf of

Professor Naim Akhtar Khan 

Academic Editor

PLOS ONE